# Malignant Brenner Tumor of the Ovary: A Systematic Review of the Literature

**DOI:** 10.3390/cancers16061106

**Published:** 2024-03-09

**Authors:** Nektarios I. Koufopoulos, Abraham Pouliakis, Menelaos G. Samaras, Christakis Kotanidis, Ioannis Boutas, Adamantia Kontogeorgi, Dionysios Dimas, Kyparissia Sitara, Andriani Zacharatou, Argyro-Ioanna Ieronimaki, Aris Spathis, Danai Leventakou, Magda Zanelli, Ioannis S. Pateras, Ioannis G. Panayiotides, Andrea Palicelli, John Syrios

**Affiliations:** 1Second Department of Pathology, Medical School, National and Kapodistrian University of Athens, Attikon University Hospital, 12462 Athens, Greece; apou1967@gmail.com (A.P.); menelaos.g.samaras@gmail.com (M.G.S.); xrkotan@gmail.com (C.K.); anniezacharatoumed25@gmail.com (A.Z.); anismed03@yahoo.gr (A.-I.I.); aspathis@med.uoa.gr (A.S.); danaileventakou@gmail.com (D.L.); ipateras@med.uoa.gr (I.S.P.); ioagpan@med.uoa.gr (I.G.P.); 2Breast Unit, Rea Maternity Hospital, 17564 Athens, Greece; drboutas@gmail.com; 33rd Department of Obstetrics and Gynecology, National and Kapodistrian University of Athens, Attikon University Hospital, 12462 Athens, Greece; ad.kontogewrgi@gmail.com; 4Breast Unit, Athens Medical Center, Psychiko Clinic, 11525 Athens, Greece; dionysis.dimas@gmail.com; 5Department of Internal Medicine, “Elpis” General Hospital of Athens, 11522 Athens, Greece; kdsitara@gmail.com; 6Pathology Unit, Azienda USL-IRCCS di Reggio Emilia, 42123 Reggio Emilia, Italy; magda.zanelli@ausl.re.it; 7Second Department of Medical Oncology, Mitera Hospital, 15123 Athens, Greece; syriosi@yahoo.gr

**Keywords:** malignant, Brenner tumor, ovarian carcinoma, lymphadenectomy, prognosis, outcome

## Abstract

**Simple Summary:**

Malignant Brenner tumors are rare ovarian neoplasms. Our aim is to provide insights concerning this rare entity. We reviewed 115 cases reported in the English literature until 15 September 2023, and analyzed the available demographic, clinical, and pathologic data. We also described the treatment modalities. A comparison of the available data showed that patients treated with lymph node dissection had a better disease-related survival rate. Disease recurrence was associated with tumor stage with marginal statistical significance and was more frequent in patients with ascites and those with abnormal CA-125 levels. Larger series with treatment details and long term follow-up data are needed to define the optimal management for this uncommon entity.

**Abstract:**

Background: Malignant Brenner tumors are rare ovarian tumors, accounting for less than 1% of malignant ovarian neoplasms. The aim of this manuscript is to systematically review the current literature concerning malignant Brenner tumors. Methods: We searched three medical databases (PubMed, Scopus, and Web of Science) for relevant articles published until 15 September 2023. Results: After applying inclusion and exclusion criteria, 48 manuscripts describing 115 cases were included in this study from the English literature. Conclusions: We analyzed the demographic, clinical, pathological, and oncological characteristics of 115 patients with malignant Brenner tumors. The statistical analysis showed that recurrence was marginally statistically significantly related to tumor stage and was more common in patients with ascites and in women with abnormal CA-125 levels; patients that were treated with lymphadenectomy had better disease-specific survival.

## 1. Introduction

Brenner tumors are an uncommon subtype of epithelial neoplasms, accounting for less than 5% of ovarian tumors [1]. They are usually unilateral and have a propensity for postmenopausal women; they are commonly asymptomatic and incidental due to their small size, but patients sometimes experience symptoms such as pain or a palpable mass [2].

The origin of these tumors is unknown. A number of them may derive from fallopian tube epithelium or Walthard nests [3], while when rarely associated with teratomas, they may originate from germ cells [1]. MacNoughton-Jones first described Brenner tumors in 1898, whereas in 1907, Fritz Brenner published the article “Das oophoroma folliculare” [4], considering them a variant of the granulosa cell tumor [5]. This neoplasm was first called a Brenner tumor by Meyer in 1932 [6]. Von Numers was the first to describe a malignant Brenner tumor (MBT) in 1945 [7].

Brenner tumors are classified into benign, borderline, and malignant variants, with benign being the most common. Borderline variants are infrequent (less than 5% of all cases), and MBTs are extremely rare, with less than 150 cases reported in the English literature. Histologically, MBTs are composed of atypical transitional/urothelial-type cells that occasionally display focal squamous differentiation. By definition, they show stromal invasion, usually with a desmoplastic stromal response, and are associated with a benign and/or borderline element [8].

This study aims to review MBTs’ clinical, pathological, diagnostic, molecular, and treatment features, focusing on differential diagnosis.

## 2. Materials and Methods

### 2.1. Systematic Review

The systematic review of the literature was performed according to the “Preferred Reporting Items for Systematic Reviews and Meta-Analyses” (PRISMA) guidelines (http://www.prismastatement.org/; accessed on 15 September 2023) (Figure 1) to identify published manuscripts of malignant ovarian Brenner tumors.

Our retrospective observational study search was conducted through the **PICO** process:**P**opulation: Women with a diagnosis of MBT;**I**ntervention: Surgical treatment of the primary ovarian tumor;**C**omparison: None;**O**utcome: Patient treatment, follow-up.

We searched for (“malignant”) AND (“Brenner”) AND (“tumor”) AND (“ovary” OR “ovarian”) in three different databases. The search yielded results on PubMed (all fields; 304 results; https://pubmed.ncbi.nlm.nih.gov, accessed on 15 September 2023), Scopus (Title/Abstract/Keywords; 515 results; https://www.scopus.com/, accessed on 15 September 2023), and Web of Science (all fields, 188 results; https://login.webofknowledge.com, accessed on 15 September 2023). We did not set any additional limitations while performing the search.

We applied the following criteria:Eligibility/inclusion criteria:
(1)**Study design:** We only included original studies and case reports describing cases of MBT.(2)**Population:** Studies involving adult patients diagnosed with MBT that provided adequate surgical and/or oncological information were included.(3)**Intervention or exposure:** We included studies that examined any treatment or intervention for MBT, including surgery, chemotherapy, radiation therapy, or targeted therapies.(4)**Outcome:** We included studies that reported on the presence or absence of disease relapse as an outcome measure.(5)**Language:** The included studies were written in the English language.Exclusion criteria:
(1)**Review articles and editorials:** We excluded narrative or systematic reviews, meta-analyses, opinion pieces, and other articles that did not present original research findings.(2)**Insufficient information:** Cases with insufficient or too much aggregated data were excluded.(3)**Uncertain diagnosis:** Cases with an uncertain/doubtful diagnosis were excluded.(4)**Histologic criteria:** Cases lacking a benign or borderline Brenner component were excluded.(5)**Language:** Manuscripts in languages other than English were excluded.

Three authors (I.B., D.D., and K.S.) worked independently to remove duplicate papers. They also reviewed the titles and abstracts of all the search results (*n* = 1007). Any disagreement was resolved by consensus. After applying eligibility and exclusion criteria, 48 manuscripts describing 115 cases of MBT were included in this review (Table 1 and Appendix A).

### 2.2. Statistical Analysis

Statistical analysis was performed via the SAS for Windows 9.4 software platform (SAS Institute Inc., Cary, NC, USA). Descriptive values were expressed as the mean ± standard deviation (SD) and, when no normality was confirmed (via the Shapiro–Wilk test), as median value, 1st (Q1) and 3rd (Q3) quartile values, respectively. For categorical data we reported the appearance frequency and the relevant percentages.

Comparisons between groups for the qualitative parameters were made using the chi-square test. For the numerical data (such as a woman’s age), normality was not possible to ensure, therefore, non-parametric tests were applied, specifically the Kruskal–Wallis test.

Furthermore, we estimated survival time using the Kaplan–Meier method; we considered that the follow-up time reported in the studies was equal to the survival time for those women that died from the disease, while in all other cases, the follow-up time was considered as the time point for censored cases. Additional tests for factors that could affect survival time were performed using the log-rank method.

The significance level (α) was set to 0.05 for all statistical tests; thus, a statistically significant difference between compared groups was when *p* < 0.05 and all tests were two sided.

## 3. Results

### 3.1. Demographic and Clinical Data

The publication years ranged from 1956 to 2023. The age in 108/115 (93.9%) cases [2,8,9,10,11,12,14,15,16,17,18,19,20,21,22,23,24,25,26,27,28,29,30,31,32,33,34,35,36,37,38,39,40,41,42,43,44,45,46,47,48,49,50,51,52,53] was reported. Specifically, the mean age at presentation was 59 ± 13 years, ranging from 22 to 87 years. Presenting symptoms were reported in 113/115 (98.3%) patients [2,8,9,10,11,12,13,14,15,16,17,18,19,20,21,22,23,24,25,26,27,28,29,30,31,33,34,35,36,37,38,39,40,41,42,43,44,45,46,47,48,49,50,51,52,53]. The most common presenting symptom was abdominal pain, which was present in 42/113 patients (37.1%) [2,13,19,22,26,27,30,36,37,38,39,40,41,45,47,48,49,51], followed by adnexal mass (15/113, 13.3%) [13,24,51], abdominal/pelvic mass (15/113, 13.3%) [24,31,35,36,40,42,43,44,48,53], abdominal distention (16/113, 14.1%) [13,14,34,36,40,43,48,52], vaginal bleeding (15/113, 13.3%) [10,13,21,24,25,27,38,40,43,48], weight loss (8/113, 7.1%) [20,25,26,31,43,44,53], abnormal uterine bleeding (6/113, 5.3%) [10,11,14,39,45], and nausea and/or vomiting (6/113, 5.3%) [10,13,47]. Other symptoms included diarrhea (2/113, 1.8%) [15,18], constipation (2/113, 1.8%) [44,47], hematuresis (1/113, 0.9%) [40] and acute urinary retention (1/113, 0.9%) [17]. Ascites was present in 33/113 (29.2%) cases [9,13,20,24,25,26,27,29,31,34,36,38,39,40,42,43,44,46,48,53]. The patient presented by Baizabal-Carvallo et al., had a bifrontal headache, tinnitus, blurred vision, and dizziness due to dural metastasis [33]. Each of these symptoms occurred alone or in combination with other symptoms. In 7/113 (6.2%) [12,13,36,38,45,50] cases, patients were asymptomatic. Details concerning symptoms can be seen in Appendix A.

Data concerning laterality were provided in 97/115 (84.3%) cases [2,8,9,10,11,12,14,16,17,18,19,20,21,22,23,24,25,26,27,28,29,30,31,32,33,34,35,36,37,38,39,40,42,43,44,46,47,48,49,50,51,52,53]; 45/97 (46.4%) cases involved the right ovary [2,9,16,17,18,21,24,25,26,29,30,31,32,35,36,37,38,39,40,42,43,46,47,48,49,51,53], 36/97 (37.1%) cases arose from the left ovary [8,10,11,12,14,19,24,27,28,30,32,33,34,36,38,40,44,48,50,51], and 16/97 (16.5%) cases showed bilateral ovarian involvement [2,20,22,23,27,36,38,40,51,52]. Tumor size was reported in 105/115 (91.3%) cases, ranging from 2 to 30 cm, with a mean value of 12.2 cm [2,8,9,10,11,13,14,15,16,17,18,20,21,22,23,24,25,26,27,28,29,30,31,32,34,35,36,37,38,39,40,42,43,44,45,46,47,48,49,50,51,52,53]. Two manuscripts, Miles and Norris [13] and Zhang et al. [45], reported the mean value and SD; these values were used for each individual patient. There was no information regarding tumor size in 9/115 (7.8%) cases [19,32,33,40,41,51]. In a single case, the tumor size was mentioned as >10 cm [42].

CA-125 serum levels were reported in 65/115 (56.5%) cases [2,8,30,32,34,36,37,38,39,40,42,43,44,45,46,47,48,51,52]. Five reports mentioned the CA-125 level as normal without providing an exact value [3,31,50]. The mean value was 202.69 U/mL, ranging from 4 to 4073.3 U/mL). Details showing patients’ demographic, treatment, and outcome characteristics are presented in Table 2.

Staging was performed in 100/115 (86.9%) cases [2,8,10,13,17,18,20,21,22,23,24,26,27,28,29,30,31,32,33,34,36,38,39,40,42,44,45,47,48,49,50,51,52,53]. Stage I disease was assigned to 50/100 (50%) patients [8,10,13,21,23,24,28,30,32,34,36,38,40,45,48,49,50,51], stage II to 7/100 (7%) patients [38,42,45,48,51], stage III to 32/100 (32%) patients [2,13,18,22,26,27,29,30,36,38,39,40,45,47,48,51,52,53], and stage IV to 11/100 (11%) patients [17,20,31,33,36,38,40,44,45]. One patient was not staged due to her poor medical status [35]. Staging was not mentioned in 14/115 (12.1%) cases [9,11,12,14,15,16,19,25,32,37,41,43,46]. The details of the staging are presented in Appendix A.

### 3.2. Diagnosis

The diagnosis of MBT, according to the latest edition of the WHO diagnostic criteria (5th edition, 2020) [1], requires the presence of invasive urothelial-like carcinoma and the presence of a benign and/or borderline Brenner tumor component. The cases included in this review satisfied these diagnostic criteria. Immunohistochemically, MBTs were positive for PAX-8 (1/3, 33%) [42,49,52], CK7 (6/6, 100%) [42,43,44,49,52], Uroplakin III (1/2, 50%) [42,52], GATA-3 (4/4, 100%) [42,43,49,50], p63 (6/6, 100%) [42,43,44,49,50,52], and negative for WT-1 (0/2, 0%) [43,52]. Some authors have described some morphologic variants of MBT. St. Pierre-Robson et al., published three cases with an unusual pattern of invasion without a desmoplastic response [8]. McGinn et al., reported two cases of a possibly novel variant of the Brenner tumor; these neoplasms consisted of a benign Brenner component associated with a low-grade basaloid carcinoma [50].

### 3.3. Surgical Management

Information regarding surgical treatment was mentioned in 110/115 (95.6%) cases [2,5,8,10,11,12,13,14,15,16,17,18,19,20,21,22,23,24,25,26,27,28,29,30,31,32,33,34,35,36,37,38,39,40,41,42,43,44,45,46,47,49,50,51,52,53]. A woman with stage IV disease did not receive surgical treatment [33]. The majority of patients (88/109, 80.7%) underwent hysterectomy and bilateral salpingo-oophorectomy (HBSO) [2,8,10,12,13,15,17,18,19,20,21,23,24,25,26,27,28,29,30,32,34,36,37,38,39,40,41,43,44,45,46,47,49,51,52,53]. The rest of the patients were treated with other procedures, such as hysterectomy and right salpingo-oophorectomy (1/109, 0.9%) [38], bilateral salpingo-oophorectomy (BSO) (10/109, 9.1%) [13,16,22,27,31,43,45,50], left salpingo-oophorectomy (5/109, 4.5%) [8,11,14,38,50], right salpingo-oophorectomy (2/109, 1.8%) [30,42], or right oophorectomy (2/109, 1.8%) [35,40]. In 2/109 (1.8%) cases [13], the procedure was salpingo-oophorectomy without mentioning the side. Omentectomy was performed in addition to HBSO or BSO in 64/109 (58.7%) patients [2,8,22,25,26,27,30,34,36,37,38,39,40,41,43,44,45,51,52,53]. Other procedures included omental biopsy/sampling (4/109, 3.6%) [8,20,27,49], excision of mesenteric nodules (1/109, 0.9%) [17], resection of bladder-involved focus (1/109, 0.9%) [40], splenectomy (1/109, 0,9%) [2], right hemicolectomy (1/109, 0.9%) [47], and appendectomy (23/109, 21.1%) [2,34,38,39,40,51]. Lymph node dissection was performed in 41/109 (37.6%) [2,36,38,39,42,43,45,51,52,53] and lymph node biopsy in 1/109 (0.9%) [37] of the cases. The applied surgical approach is detailed in Appendix A.

### 3.4. Adjuvant Therapy

Information concerning adjuvant treatment was reported in 96/115 (83.5%) of cases [2,5,10,13,15,16,17,18,19,21,22,23,27,30,31,33,35,36,37,38,39,40,41,42,43,44,45,46,47,48,49,50,51,52,53]. Adjuvant therapy was not administered to 27/96 (28.1%) patients [10,13,15,17,18,19,23,31,33,38,40,43,45,47,49,50,51]. Most of them had stage I disease. In one case, the patient refused adjuvant therapy [19]. In two cases with stage IV disease, the reasons were the patient’s poor status in the first [31] and that the patient died a few hours after surgery in the second [33]. Radiotherapy was offered alone in 3/69 (4.3%) [10,13,16] or in combination with chemotherapy in 4/69 (5.8%) patents [21,38,51,52]. Chemotherapy was administered in 63/93 (67.7%) patients [2,21,22,27,30,32,35,36,37,38,39,40,41,42,44,45,46,48,51,52,53]. The most commonly used regimen was paclitaxel-carboplatin (TC) in 41/63 (65%) of patients [2,3,38,39,42,45,46,48,51,52,53], followed by Melphalan (Alkeran) (5/63, 7.9%) [21,22,27], paclitaxel-cisplatin (3/63, 4.7%) [40], and various other drug combinations [27,30,32,37,38,40,45]. Neoadjuvant chemotherapy was administered in two cases with stage IIIb and stage IV disease, consisting of six cycles of TC and five cycles of paclitaxel-cisplatin, respectively [38,45]. 

In 33/46 (71.7%) cases with disease relapse, information concerning treatment was available [10,15,19,22,23,27,30,32,36,38,40,41,42,45,48,51], including tumor debulking surgery (6/33, 18.1%) [32,41,42,45,48,51], radiotherapy (6/33, 18.1%) alone [15,27] or in combination with surgery and/or chemotherapy [42,45,51]. In 27/33 (81.8%) patients, chemotherapy was administered [22,23,30,32,36,38,40,41,42,45,48,51]. The most common therapeutic regimen was TC used in 12/27 (48%) of cases [36,38,42,48,51] with various other combinations [22,23,30,38,40,41,42,45,48,51]. Details of adjuvant treatment for each patient are presented in Appendix A.

### 3.5. Molecular Findings

Two cases were tested for *BRCA1/2* mutations [41,49]. A *BRCA-2* pathogenic mutation was present in the case reported by Toboni et al. [41]. No other information was provided.

### 3.6. Follow-up and Survival

Follow-up data were available in 106/115 (92.1%) cases [2,8,9,10,11,13,15,16,17,18,19,21,22,23,24,25,27,29,30,31,32,33,34,35,36,37,38,39,40,41,42,43,44,45,47,48,49,50,51,52,53]; 53/106 (50%) patients were alive without evidence of the disease [2,8,10,11,13,21,24,25,32,34,36,37,38,39,40,42,43,44,45,49,50,51,52,53], 10/106 (9.4%) were alive with the disease [27,31,38,41,45,48], 30/106 (28.3%) succumbed to the disease [9,10,13,15,16,17,18,19,22,23,27,29,30,32,33,35,36,38,40,45,47,51], 6/106 (5.7%) died of other causes [13,24,38,40], and 5/106 (4.7%) were lost at follow-up [38,45,48].

Follow-up time was specified in 102/115 (88.7%) cases [2,8,9,10,11,13,15,16,17,18,19,21,22,27,29,30,31,32,33,34,35,36,37,38,39,40,42,43,45,48,49,50,51,52,53], ranging from 1 to 173 months (mean: 40.1 months). For all except one woman, information on the outcome was available, thus survival curves were possible to construct; the mean survival time for all patients was estimated with the Kaplan–Meier approach at 80.9 months (standard error: 5.5 months) (Figure 2). 

Relapse information was available in 104/115 (90.4%) cases [2,10,11,13,15,16,17,18,19,21,22,23,24,25,27,29,30,32,33,34,35,36,37,38,39,40,41,42,43,44,45,46,47,48,49,50,51,52,53]; 46/104 (43.2%) patients had one or more relapses [10,13,15,19,22,23,27,29,30,32,35,36,38,40,41,42,45,48,50,51], while there was no disease relapse in 59/104 (56.8%) cases [2,10,11,13,16,17,18,21,24,25,32,33,34,36,37,38,39,40,43,44,45,46,47,49,50,51,52,53]. The median time to relapse was 13 months (Q1–Q3: 9–36 months), and the mean time was 25.5 months (range 3–116 months). Regarding the relapse site, there was available information for 27/46 (60%) patients [10,15,17,19,20,23,27,29,30,32,38,42,48,50,51]. The most common sites were the liver in 11/27 (40.7%) [10,20,38,40,48,51], lymph nodes in 6/27 (22.2%) [22,30,38,42,51], bone in 5/27 (18.5%) [15,27,32,38,50], lung in 4/27 (14.8%) [38,40,50,51], peritoneum in 5/27 (18.5%) [10,19,27,30,48], and the omentum in 4/27 (14.8%) of the cases [20,27,30].

### 3.7. Results of Inferential Statistical Analysis

The available data allowed for the performance of inferential statistics and the extraction of possible relations. A possible role of the tumor side (left or right) and the development of ascites was not possible to confirm (*p* = 0.1165). We furthermore studied all collected data for their role in recurrence, with the results being summarized in Table 3. 

Age, tumor size, tumor location (left or right), and the administration of adjuvant therapy did not have any statistically significant impact on subsequent recurrence. CA-125 was higher in women with recurrence (median: 91.7 Q1–Q3: 43–273.4, vs. median: 27 Q1:Q3: 13–184.2, *p* = 0.1164). When considering CA-125 levels as normal/abnormal (using 35 U/mL as a cut-off the value), the percentage of women who had normal CA-125 levels and still recurred was only 29.63%, while it was 70.37% for women without recurrence. The correlation of CA125 to disease recurrence was marginally significant (*p* = 0.0522) without enough statistical power to make a definitive statement about it. Moreover, it was observed that in women with recurrence, ascites was more common (38.1% vs. 22.5%, *p* = 0.1033). Clearly, stage was a decisive factor for recurrence (see Table 3), since 24.4% of the women with stage I had a recurrence, while the percentage was more than 60% for disease at stage II–IV (*p* = 0.0018).

The tumor side (left, right, or bilateral) had no role in patient survival time (log-rank *p* = 0.9378; Figure 3 highlights relevant survival curves and the number of women at risk). 

In contrast, an abnormal CA-125 level was linked to lower survival (Figure 3, *p* = 0.0476), with a mean survival of 29 months (Q1–Q3: 20–64 months) and 47 months (Q1–Q3: 24–96 months) for abnormal and normal CA-125 status, respectively. Similarly, women with tumors at stage I experienced better survival than women at stages higher than I (Figure 3, *p* = 0.0057); specifically, the median survival was 53 months (Q1–Q3: 24–94 months) for stage I cases and 39 months (Q1–Q3: 20–78 months) for tumors at stage higher than I, respectively. Furthermore, ascites was not an important factor for lower survival (*p* = 0.8735). Finally, patients with lymph node dissection (LND), had better survival than patients without LND (*p* = 0.0131); specifically, the median survival for the 34 women in whom LND was performed was 117 months, and for the women without LND, it was 69 months.

## 4. Discussion

Ovarian cancer is the fifth most common cause of cancer-related death from gynecological carcinomas [54,55]. Due to their rarity, MBTs comprise only a small fraction of these tumors. To our knowledge, this study is the first to review the literature systematically. In 1988, Austin and Morris first recognized that a subgroup of MBTs lacking a benign Brenner component represented, in fact, high-grade ovarian serous carcinomas with a transitional architectural pattern [56]. To ensure that we did not include such cases, we included, for cases reported before 1988, only invasive tumors associated with a benign and/or borderline Brenner component. 

In our study, the mean age of patients presenting with MBT is 59 years. In comparison, a previous study reported the mean age of patients to be 65 years [57]. For other histotypes, the age of presentation ranges from 55 years for mucinous and endometrioid carcinoma, 56 years for clear-cell carcinoma, and 65 years for serous carcinoma [1]. MBTs tend to present at a lower stage compared to serous carcinoma [1]. The symptoms of MBT are similar to those of other epithelial ovarian carcinomas. The most common symptoms reported were abdominal pain, adnexal, abdominal or pelvic mass, abdominal distention, and vaginal bleeding. According to the literature, ascites is present in <10% of MBT cases. Our study reveals a much higher (28.9%) percentage. MBTs have no specific ultrasound or MRI findings [58,59].

The inferential statistical analysis performed in our study showed that disease stage I is associated with a statistically significant lower percentage of disease recurrence compared to stages II-IV. Also, disease recurrence is more commonly related to the presence of ascites and elevated CA-125 levels. Furthermore, the analysis showed a relation between higher CA-125 levels and a stage higher than I with decreased survival. In contrast to the study by Nasioudis et al., our analysis showed that patients treated with LND had a better survival rate [57].

The first step in correctly managing every malignancy is a precise diagnosis. The differential diagnosis of MBT includes high-grade ovarian serous carcinoma with a transitional architectural pattern, primary squamous cell carcinoma (SqCC), SqCC arising in a mature cystic teratoma, endometrioid borderline tumor, endometrioid carcinoma, metastatic SqCC, and metastatic urothelial carcinoma.

High-grade ovarian serous carcinoma with a transitional architectural pattern shows areas of conventional high-grade serous carcinoma with high-grade nuclear atypia, prominent nucleoli, and significant pleomorphism. It lacks a benign Brenner component, and, immunohistochemically, it is positive for WT-1 and estrogen receptors [60].

Primary ovarian SqCC usually shows keratinization and high-grade nuclear features, lacking a benign Brenner component; it may arise from a mature teratoma [61,62]. Endometrioid borderline tumors and endometrioid carcinoma show at least partially endometrioid-type glands and are immunohistochemically positive for ER; they are frequently related to endometriosis.

In the differential diagnosis of metastatic tumors (either SqCC or urothelial carcinoma), knowledge of the previous clinical history is of great importance. Furthermore, metastatic tumors tend to be bilateral, displaying a multinodular growth pattern and lacking a benign Brenner component. 

For instance, metastatic SqCC also does not show a papillary architecture. The summary of essential clinical, histologic, and immunohistochemical features for the distinction of the entities mentioned above is shown in Table 4 and Table 5.

Concerning the molecular findings in MBTs, the most common are inactivating mutations in the *CDKN2A* and *CDKN2B* loci encoding the cyclin-dependent kinase inhibitors p16INK4a and p15INK4b, respectively, followed by activating mutations in *FGFR3* and *PIK3CA* [63]. Notably, the p53 signaling was frequently disrupted in MBTs. The amplification of murine double minute 2 (*MDM2*)—encoding an E3 ubiquitin ligase that counteracts p53 suppressor activity—was a frequent event [63]. Only a few cases harbored *TP53* truncating and missense mutations, which were shown in a mutually exclusive pattern with *MDM2* amplification [64]. Interestingly, *MDM2* amplification or *TP53* mutations were mainly present in *FGFR3* wild-type cases [63]. Wang et al., reported amplification of *MDM2* and *CCND1* (encoding Cyclin D1), and loss of CDKN2A and CDKN2B in one case of MBT [48]. Also, MBTs lack *TERT* promoter mutations, commonly found in urothelial carcinoma [65,66]. Genomic alterations in genes involved in the homologous recombination deficiency (HRD) pathway were rare; Lin et al., revealed homozygous inactivating mutations only in BAP1 in rare cases [63]. A pathogenic *BRCA2* mutation was found in the case presented by Toboni et al. [41]. Overall, it seems that MBT has unique molecular features among gynecological malignancies. In addition, previous data revealed that the FGFR3 and MDM2/P53 pathways, along with CDKN2A/B loss, play a key role in the pathogenesis of MBT. However, as MBT is rare, additional studies are required to shed light on the molecular events driving this entity. A summary of the molecular alterations is presented in Appendix A.

Surgery is the basis of MBT treatment. The majority of patients in our review were treated with HBSO, with or without omentectomy, appendectomy, and lymph node dissection.

The role of adjuvant chemotherapy has yet to be defined. In early stage disease, the benefit of chemotherapy is not clear. For instance, Gezginc et al., reported that patients of stages IA and IB could be followed up, and Han et al., spared patients of stage IA disease from chemotherapy [45]. It is reasonable, therefore, to discuss with the patient the pros and cons and potentially offer adjuvant chemotherapy to those with stage IC and higher disease due to increased recurrence risk. 

Literature shows that most clinicians have been using alkylating agents (such as cisplatin, cyclophosphamide, and melphalan), tumor antibiotics (mitomycin C and doxorubicin), and, importantly, taxanes (mainly docetaxel and paclitaxel) in treating MBT, either in the adjuvant or metastatic setting [2,27,30,36,37,38,39,40,41,42,48,51,52,53]. Since 2012, the combination of platinum with taxane has been gaining rising acceptance among clinicians, and carboplatin with paclitaxel is currently the most used regimen [2,36,38,39,40,41,42,44,45,48,51,52,53,67]. This is in line with the international guidelines, which suggest that patients with high grade histology should be treated with six cycles of carboplatin and paclitaxel chemotherapy.

Importantly, antiangiogenic factors increase the progression free survival of patients with locally advanced and metastatic, high-grade epithelial ovarian cancer; however, patients with MBT were not included in these clinical trials [68,69]. Lang et al., reported clinical benefit with the addition of Bevacizumab in a patient with recurrent MBT [42]. 

Due to the rarity of the disease, patients with recurrent or metastatic disease should be encouraged to undergo a genetic next-generation sequencing analysis of the tumor. This may shed light on the pathogenesis of this malignancy and allow for a treatment approach tailored to the patient.

Data on the role of radiotherapy are lacking in the literature. Only a few cases are reported, receiving radiotherapy as part of their adjuvant treatment [21,51] and in the case of recurrence [27,42,45,51]. The use of radiotherapy cannot be advocated, particularly in early stage disease; it is reasonable, however, to consider targeted radiotherapy for symptom control.

Besides, the low incidence of this disease does not permit clinicians to carry out randomized clinical trials. Treatment protocols are therefore based on a case-by-case experience. It is therefore highly recommended that these cases be discussed in multidisciplinary team boards and published to accumulate clinical evidence.

## 5. Conclusions

In the present manuscript, we have collected data presenting a systematic review of MBTs’, presenting their demographic, clinical, pathological, molecular, and treatment characteristics, with a special focus on the differential diagnosis. To our knowledge, this is the first study to systematically review the characteristics of these tumors. More multicentric studies reporting in detail treatment modalities and long-term follow-up are needed to define the optimal management for this rare entity.

## Figures and Tables

**Figure 1 cancers-16-01106-f001:**
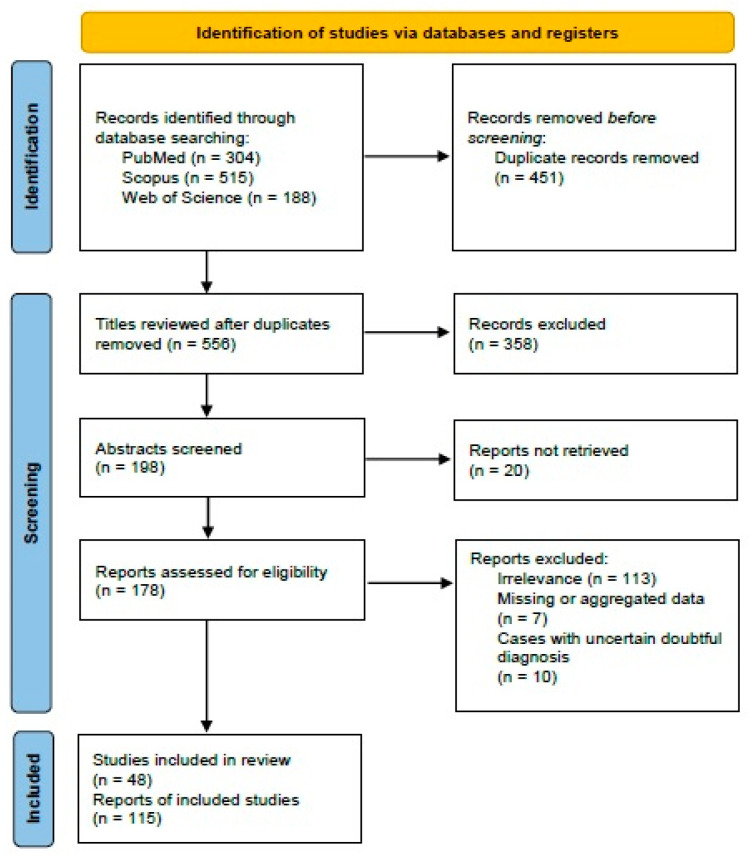
PRISMA 2020 flowchart showing the search strategy, excluded studies, and finally included reports.

**Figure 2 cancers-16-01106-f002:**
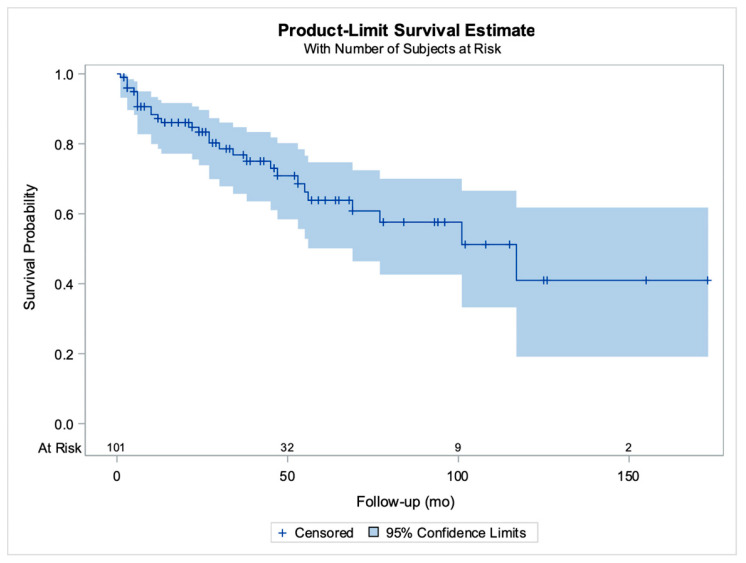
Kaplan–Meier curves for patient survival. The horizontal axis shows the follow-up period in months and the number of patients at risk for various time points, vertical lines correspond to censored cases (previously unpublished original photo).

**Figure 3 cancers-16-01106-f003:**
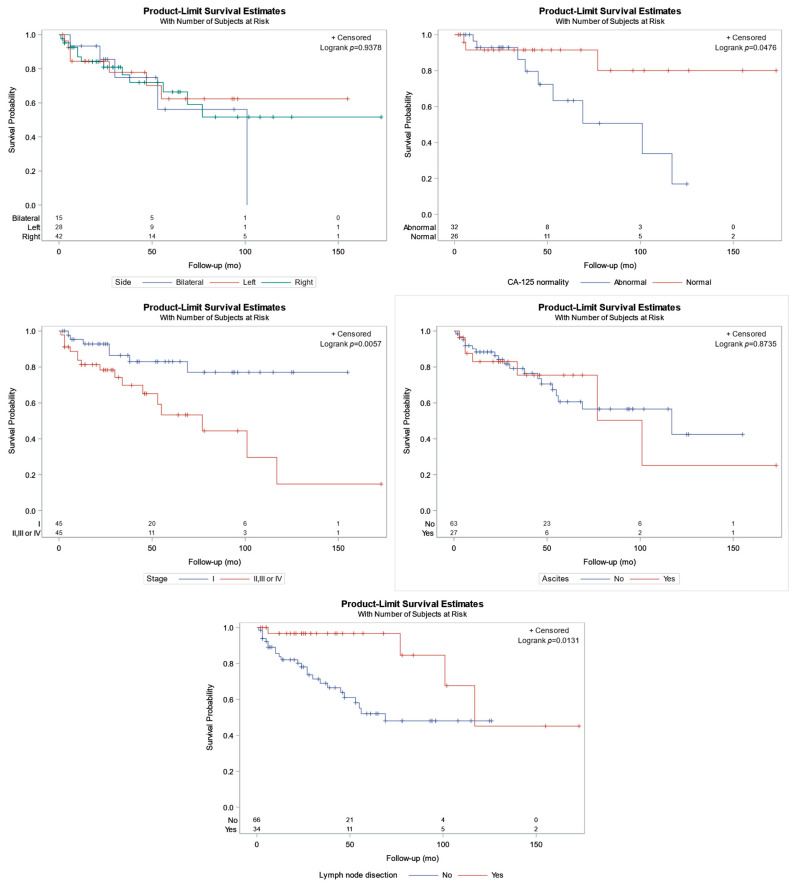
Kaplan–Meier curves for patient survival in relation to the upper left: bilaterality; upper right: CA-125 characterization (normal/abnormal); middle left: stage (I vs. II, II, or IV); middle right: ascites; bottom: lymph node dissection (previously unpublished original photo).

**Table 1 cancers-16-01106-t001:** Clinic-pathologic and treatment features of the cases of malignant Brenner tumors.

Authors	Year	Age	Presentation	Side	Tumor Size (cm)	Stage	CA-125 (U/mL)	Ascites	Surgery	Adjuvant Therapy	Recurrence	Interval to Recurrence (mo)	Second Line Therapy	Follow-Up (mo)	Outcome
Mackinlay [9]	1956	64	Pain	Right	15	NM	NM	Yes	NM	NM	NM	NM	NM	6	DOD
Abel [10]	1957	48	AUB, pain	Left	5	Ib	ΝΜ	No	HBSO	RT	No	NA	NA	2	ANED
Abel [10]	1957	62	Vaginal bleeding, pain, nausea, vomiting	Left	25	Ia	ΝΜ	No	HBSO	None	Yes	27	No	27	DOD
Reel [11]	1958	84	AUB, lower abd. enlargement, pressure symptoms	Left	18	NM	ΝΜ	No	LSO	NM	No	NA	NA	24	ANED
Marshall [12]	1970	69	Asymptomatic	Left	10	NM	ΝΜ	No	HBSO	NM	NM	NA	NA	NM	NM
Miles and Norris [13]	1972	*	**	NM	***	I	ΝΜ	****	HBSO	None	No	NA	NA	5	ANED
Miles and Norris [13]	1972	*	**	NM	***	I	ΝΜ	****	SO	None	No	NA	NA	42	ANED
Miles and Norris [13]	1972	*	**	NM	***	I	ΝΜ	****	HBSO	None	No	NA	NA	25	DOC
Miles and Norris [13]	1972	*	**	NM	***	I	ΝΜ	****	SO	None	No	NA	NA	78	DOC
Miles and Norris [13]	1972	*	**	NM	***	III	ΝΜ	****	BSO	None	Yes	NM	NA	3	DOD
Miles and Norris [13]	1972	*	**	NM	***	I	ΝΜ	****	HBSO	None	Yes	NM	NA	13	DOD
Miles and Norris [13]	1972	*	**	NM	***	I	ΝΜ	****	HBSO	RT	Yes	NM	NA	27	DOD
Toriumi and Ijima [14]	1973	55	Uterine bleeding, abd. distension	Left	20	NM	ΝΜ	No	LSO	NM	NM	NM	NM	NM	NM
Hull and Campbell [15]	1973	52	Abd. enlargement, abd. discomfort, diarrhea	Left	15	NM	NM	No	HBSO	None	Yes	36	RT	47	DOD
Pratt-Thomas et al. [16]	1976	59	Dyspnea, right pleural effusion	Right	14	NM	NM	No	BSO	RT	Yes	NM	NM	56	DOD
Beck et al. [17]	1977	67	Acute urinary retention	Right	14	IV	NM	No	HBSO, excision of mesenteric nodules	None	No	NA	NA	10	DOD
Shafeek et al. [18]	1978	23	Right hypochondrium swelling, diarrhea	Right	9	IIIC	NM	No	HBSO	None	No	NA	NA	1	DOD
Chiarelli et al. [19]	1978	67	Abd. pain	Left	NM	NM	NM	No	HBSO	None (patient refused)	Yes	3	Inoperable tumor-biopsy	6	DOD
Hayden [20]	1981	68	Increased abd. girth, WT loss	Bilateral	6.5–6.5	IV	NM	Yes	HBSO, omental biopsy	NM	NM	NM	NM	NM	NM
Magrina et al. [21]	1982	63	Vaginal bleeding	Right	10	Ic1	NM	No	HBSO	CHT; RT	No	NA	NA	53	ANED
Haid et al. [22]	1983	78	Abd. pain	Bilateral	10 + 5	IIIc	NM	No	BSO, OM	CHT	Yes	9	R1: CHT R2: CHT	22	DOD
Chen [23]	1984	73	Pelvic discomfort.	Bilateral	12 + 8	Ia	NM	No	HBSO	None	Yes	4	CHT	6	DOD
Roth and Czernobilsky [24]	1985	42	Adnexal mass	Right	14	Ia1	NM	#	HBSO	NM	No	NA	NA	33	ANED
Roth and Czernobilsky [24]	1985	74	Abd. mass	Left	13.5	Ia1	NM	#	HBSO	NM	No	NA	NA	7	ANED
Roth and Czernobilsky [24]	1985	76	Vaginal bleeding, abd. mass	Right	17	Ic	NM	#	HBSO	NM	No	NA	NA	108	DOC
Roth and Czernobilsky [24]	1985	65	Adnexal mass	Right	9	Ia2	NM	#	HBSO	NM	No	NA	NA	65	ANED
Seldenrijk et al. [25]	1986	79	Vaginal bleeding, abd. mass, WT loss	Right	14	NM	NM	Yes	HBSO, OM	NM	No	NA	NA	7	ANED
Hayashi et al. [26]	1987	67	Abd. pain, WT loss	Right	10	III	NM	Yes	HBSO, OM	NM	NM	NA	NA	NM	NM
Chen and Hoffman [27]	1988	59	Abd. pain, tenderness	Left	6	IIIc	NM	No	BSO, OM	CHT	Yes	78	NM	78	AWD
Chen and Hoffman [27]	1988	72	Vaginal bleeding	Left	15	IIIc	NM	Yes	HBSO, omental biopsy	CHT	Yes	69	NM	69	AWD
Chen and Hoffman [27]	1988	69	Abd. pain	Bilateral	14 + 6	IIIc	NM	No	BSO	CHT	Yes	5 and 29	RT	30	DOD
Thirumavalavan et al. [28]	1992	63	Abd. discomfort and swelling	Left	20	Ia	NM	No	HBSO	NM	NM	NM	NM	NM	NM
Joh et al. [29]	1995	79	Abd. swelling, vaginal bleeding	Right	11	III	NM	Yes	HBSO	NM	Yes	NM	NM	3	DOD
Kataoka et al. [30]	1995	67	Abd. pain, tenderness	Left	15	IIIc	NM	No	HBSO, OM	CHT	Yes	9	CHT	55	DOD
Kataoka et al. [30]	1995	51	Abd. tenderness	Right	13	Ia	120	No	RSO	CHT	Yes	12 and 60	R1:CHT R2:CHT	69	DOD
Ahr et al. [31]	1997	77	Pelvic mass, WT loss	Right	5	IV	Normal	Yes	BSO	None (poor patient status)	NM	NM	NM	3	AWD
Yamamoto et al. [32]	1999	55	NM	Left	NM	Ia	265.3	No	HBSO	NM	Yes	42 and 50	OM, liver tumor, retroperitoneal LND bilateral ovarian vessel dissection; CHT	NM	DOD
Yamamoto et al. [32]	1999	70	Discomfort	Right	15	NM	NM	No	HBSO	CHT	No	NA	NA	20	ANED
Baizabal-Carvallo et al. [33]	2010	56	Bifrontal headache, tinnitus, blurred vision and dizziness	Left	NM	IV	NM	No	No	None	NA	NA	NA	3	DOD
Dris et al. [34]	2010	77	Abd. distension and pelvic pain	Left	16	Ic	294	Yes	HBSO, OM, AP	NM	No	NA	NA	3	ANED
Roth et al. [35]	2012	85	Abd. mass	Right	9	not staged	NM	No	RO	CHT	Yes	NM	NM	24	DOD
Gezginç et al. [36]	2012	55	Abd. distention	Left	5	IIIc	27	Yes	HBSO, OM, LND	CHT	No	NA	NA	6	DOD
Gezginç et al. [36]	2012	55	Abd. pain	Right	8	Ic	Normal	No	HBSO, OM, LND	CHT	No	NA	NA	84	ANED
Gezginç et al. [36]	2012	66	Bleeding	Left	6.5	IIIc	9.6	No	HBSO, OM, LND	CHT	Yes	11	CHT	68	ANED
Gezginç et al. [36]	2012	49	Abd. pain	Bilateral	4.5 + 3	Ib	Normal	No	HBSO, OM, LND	None	Yes	12	CHT	52	ANED
Gezginç et al. [36]	2012	43	Abd. pain	Bilateral	8 + 2	Ia	12.5	No	HBSO, OM, LND	-	No	NA	NA	57	ANED
Gezginç et al. [36]	2012	65	Abd. pain	Left	15	IIIc	208	Yes	HBSO, OM, LND	CHT	Yes	36	CHT	46	ANED
Gezginç et al. [36]	2012	79	Abd. distention	Left	7	IV	75	Yes	HBSO, OM, LND	CHT	Yes	5	CHT	5	ANED
Gezginç et al. [36]	2012	46	Pelvic mass	Right	16.5	Ic	25	Yes	HBSO, OM, LND	CHT	Yes	34	CHT	43	ANED
Gezginç et al. [36]	2012	46	Menstrual irregularity	Bilateral	15.5 + 5	Ib	Normal	No	HBSO, OM, LND	-	No	NA	NA	38	ANED
Gezginç et al. [36]	2012	46	Asymptomatic	Right	6.5	IV	67	No	HBSO, OM, LND	CHT	Yes	21	CHT	29	ANED
Gezginç et al. [36]	2012	75	Abd. pain	Bilateral	12 + 10	IIIc	448	Yes	HBSO, OM, LND	CHT	No	NA	NA	26	ANED
Gezginç et al. [36]	2012	49	Pelvic mass	Bilateral	10 + 11	IIIc	135	Yes	HBSO, OM, LND	CHT	No	NA	NA	25	ANED
Gezginç et al. [36]	2012	50	Abd. pain	Bilateral	4 + 2.5	IIIc	44	No	HBSO, OM, LND	CHT	Yes	18	CHT	20	ANED
Verma et al. [37]	2012	60	Abd. pain	Right	8	NM	4073.3	No	HBSO, OM, LN biopsy	CHT	No	NA	NA	6	ANED
St Pierre-Robson et al. [8]	2013	53	Abd. bloating	Left	7.5	Ia	NM	No	HBSO	NM	NM	NM	NM	22	ANED
St Pierre-Robson et al. [8]	2013	57	Abd. fullness	Left	13	Ia	99	No	LSO, omental sampling	NM	NM	NM	NM	24	ANED
St Pierre-Robson et al. [8]	2013	68	NM	Left	17	Ia	NM	No	HBSO, OM	NM	NM	NM	NM	NM	NM
Han et al. [38]	2014	37	Abd. pain, vaginal bleeding	Right	8	Ia	35.5	No	HRSO, OM, AP, PPLND	None	No	NA	NA	26	lost
Han et al. [38]	2014	42	Vaginal bleeding	Left	3	Ia	21.1	No	HBSO, OM, AP, PPLND	None	No	NA	NA	155	lost
Han et al. [38]	2014	59	Abd. pain	Right	2.5	IV	10.7	Yes	HBSO, OM, AP, PPLND	neoadj. CHT; CHT	Yes	9	CHT	173	ANED
Han et al. [38]	2014	52	Abd. pain	Right	10.5	IIIc	8.3	Yes	HBSO, OM, AP, PPLND	CHT	Yes	18	NM	77	DOD
Han et al. [38]	2014	61	Abd. pain	Right	13.5	Ic	23.4	Yes	HBSO, OM, AP, PPLND	CHT	No	NA	NA	21	DOC
Han et al. [38]	2014	43	Abd. pain	Bilateral	7.5 + 6.5	IIc	724	Yes	HBSO, OM, AP, PPLND	CHT; RT	Yes	9	NM	101	DOD
Han et al. [38]	2014	59	Abd. pain, mass	Right	25	Ia	13	No	HBSO, OM, AP, PPLND	None	No	NA	NA	102	ANED
Han et al. [38]	2014	68	Abd. pain, mass	Right	12.5	Ia	38.5	No	LSO	CHT	No	NA	NA	8	lost
Han et al. [38]	2014	48	Asymptomatic	Right	5.5	IIIc	4	No	HBSO, OM, AP, PPLND	CHT	Yes	13	CHT	32	AWD
Han et al. [38]	2014	61	Mass	Left	12	Ia	10.7	No	HBSO, OM, AP, PPLND	None	No	NA	NA	16	ANED
Di Donato et al. [39]	2016	46	AUB, abd. pain	Right	9	IIIc	77.8	Yes	HBSO, OM, AP, PPLND	CHT	No	NA	NA	29	ANED
Yue et al. [40]	2016	51	Abd. pain	Right	25	Ic	53.78	No	HBSO, OM, AP	CHT	No	NA	NA	38	DOD
Yue et al. [40]	2016	56	Abd. distension	Right	NM	IIIc	143	Yes	HBSO, OM, AP	CHT	Yes	13	CHT	10	DOD
Yue et al. [40]	2016	43	Pelvic mass	Right	19	IV	45.69	Yes	RO	CHT	Yes	5	CHT	34	DOD
Yue et al. [40]	2016	55	Abd. pain	Right	NM	Ic	222.4	No	HBSO, OM	CHT	Yes	22	CHT	61	ANED
Yue et al. [40]	2016	44	Abd. distension	Left	15	Ia	16.07	No	HBSO, OM, AP	CHT	No	NA	NA	5	DOD
Yue et al. [40]	2016	32	Hematuresis	Left	NM	IV	NA	No	HBSO, OM, bladder-involved focus resection	CHT	Not applicable	NA	NA	14	ANED
Yue et al. [40]	2016	46	Abd. distension	Right	11	IIIc	356.07	Yes	HBSO, OM	CHT	Yes	4	CHT	46	ANED
Yue et al. [40]	2016	48	Abd. distension, vaginal bleeding	Left	24	Ic	NA	No	HBSO, OM	CHT	No	NA	NA	93	ANED
Yue et al. [40]	2016	47	Pelvic mass	Bilateral	6 + 6	Ib	NA	No	HBSO, OM	CHT	No	NA	NA	94	DOC
Yue et al. [40]	2016	76	Vaginal bleeding, pelvic mass	Left	15	Ic	NA	No	HBSO, OM	None	No	NA	NA	94	DOC
Turgay et al. [2]	2017	49	Abd. pain, mass	Bilateral	14	IIIc	51	No	HBSO, OM, AP, PPLND, splenectomy	CHT	No	NA	NA	24	ANED
Turgay et al. [2]	2017	62	Abd. pain, mass	Right	18	IIIc	24	No	HBSO, OM, AP, PPLND	CHT	No	NA	NA	18	ANED
Toboni et al. [41]	2017	54	Gastrointestinal complaints, abd. pain	NM	NM	NM	675	No	HBSO, OM	CHT	Yes (4)	48	R1: Surgery; CHT; R2: NA; R3: NA; R4: CHT	NA	AWD
Lang et al. [42]	2017	77	Pelvic mass	Right	>10	IIc	14	Yes	RSO, PPLND	CHT	Yes	12	CHT; Surgery; RT	24	ANED
King et al. [43]	2018	58	vaginal bleeding, abd. fullness, increasing urinary pressure, and frequency	Right	25	NM	19.8	No	HBSO, OM, LND	None	No	NA	NA	2	ANED
King et al. [43]	2018	79	Pelvic mass, abd. distension, pelvic discomfort, WT loss	Right	25	NM	563.5	Yes	BSO	None	No	NA	NA	24	ANED
Agius et al. [44]	2018	70	Abd. mass, WT loss, and constipation.	Left	18	IVb	11.13	Yes	HBSO, OM	CHT	No	NA	NA	NM	ANED
Zhang et al. [45]	2019	77	AUB	NM	##	IIb	43	No	HBSO, OM, LND	CHT	Yes	116	No	117	DOD
Zhang et al. [45]	2019	58	Pelvic pressure	NM	##	Ia	12.6	No	HBSO, OM, LND	CHT	No	NA	NA	42	ANED
Zhang et al. [45]	2019	60	Abd. pain	NM	##	IIb	91.7	No	BSO, OM, LND	CHT	Yes (2)	12	R1: CHT; RT R2: CHT	12	AWD
Zhang et al. [45]	2019	67	Abd. pain	NM	##	IIa	25.4	No	BSO, OM, LND	CHT	No	NA	NA	5	ANED
Zhang et al. [45]	2019	39	Abd. pain	NM	##	IVb	494.8	No	HBSO, OM	CHT6	Yes (3)	17	R1: Surgery; RT; R2, R3: Cyberknife	45	DOD
Zhang et al. [45]	2019	70	Asymptomatic	NM	##	Ic1	10.5	No	HBSO, OM	CHT	No	NA	NA	37	ANED
Zhang et al. [45]	2019	69	AUB, abd. pain	NM	##	Ib	264	No	HBSO, OM, LND	CHT	No	NA	NA	78	ANED
Zhang et al. [45]	2019	82	Abd. pain	NM	##	IIIb	NA	No	HBSO, OM	neoadj. CHT	Yes	NM	NM	28	AWD (lost)
Zhang et al. [45]	2019	58	Pelvic pressure	NM	##	Ia	9.1	No	HBSO	None	No	NA	NA	126	ANED
Zhang et al. [45]	2019	49	Abd. pain	NM	##	Ia	10.8	No	HBSO, OM, LND	NA	NA	NA	NA	NM	lost
Toshniwal et al. [46]	2020	65	postmenopausal bleeding, abd. fullness	Right	14.1		227	Yes	HBSO	CHT	No	NA	NA	NM	NA
Singh et al. [47]	2020	62	Abd. pain, vomiting and constipation, anorexia, WT loss	Right	10.2	IIIc	184.2	No	HBSO, right hemicolectomy	None	No	NA	NA	NA	DICU
Bouhani et al. [48]	2020	73	Abd. distension	Left	15	IIc	294	Yes		CHT	Yes	9	Symptomatic treatment	14	AWD
Bouhani et al. [48]	2020	46	Abd. pain	Left	9	IIIc	490	Yes		CHT	Yes (2)	11 and 31	R1:CHT R2: CHT	39	AWD
Bouhani et al. [48]	2020	60	Pelvic mass	Right	8	IIIc	273.4	Yes		CHT	Yes (2)	11 and 18	R1: Surgery; CHT R2: symptomatic treatment	64	AWD
Bouhani et al. [48]	2020	58	Pelvic pain	Left	18	Ic	NA	Yes		CHT	Yes	59	CHT	59	lost
Wang et al. [49]	2020	71	Vaginal bleeding, abd. pain	Right	20	Ia	NM	No	HBSO, peritoneal, and omental biopsies	None	No	NA	NA	18	ANED
McGinn et al. [50]	2021	22	Asymptomatic	Left	11	Ia	Normal	No	LSO	None	Yes	50	NM	NM	NM
McGinn et al. [50]	2021	60	Asymptomatic	Left	4.5	Ia	NM	No	BSO	CHT	No	NA	NA	14	ANED
Yüksel et al. [50]	2022	75	Adnexal mass	Bilateral	3.6	IIIc	20	No	NM	CHT	No	NA	NA	47	ANED
Yüksel et al. [51]	2022	57	Adnexal mass	Left	5.5	IIa	NA	No	NM	CHT; RT	No	NA	NA	12	NA
Yüksel et al. [51]	2022	48	Adnexal mass	Left	20	Ia	9.6	No	NM	None	No	NA	NA	96	ANED
Yüksel et al. [51]	2022	37	Adnexal mass	Right	30	Ic1	12	No	NM	CHT	No	NA	NA	115	ANED
Yüksel et al. [51]	2022	49	Adnexal mass	Right	NA	IIIc	NA	No	NM	CHT	Yes	86	Surgery; CHT	96	ANED
Yüksel et al. [51]	2022	75	Adnexal mass	Right	NA	IIIc	95	No	NM	CHT	No	NA	NA	12	DOD
Yüksel et al. [51]	2022	36	Adnexal mass	Right	18	Ic3	209	No	NM	CHT	No	NA	NA	125	ANED
Yüksel et al. [51]	2022	55	Adnexal mass, abd. pain	Bilateral	15	IIIc	64	No	NM	CHT	Yes	13	CHT; RT	53	DOD
Zou et al. [52]	2022	50	Abd. distension, pain	Bilateral	15.2–6.2	IIIc	256.3	No	HBSO, OM, LND	CHT; RT	No	NA	NA	12	ANED
Kurniadi et al. [53]	2023	39	Abd. mass, WT loss	Right	25	IIIa	NA	Yes	HBSO, OM, LND	CHT	No	NA	NA	3	ANED

Abd.: Abdominal; ANED: alive with no evidence of disease; AP; appendectomy; AUB: abnormal uterine bleeding; AWD: alive with disease; CHT: chemotherapy; DOC: died of other cause; DOD: died of disease; DICU: died in intensive care unit; HBSO: hysterectomy and bilateral salpingo-oophorectomy; OM: omentectomy; LND: lymph node dissection; NA: not applicable; NM: not mentioned; PPLND: pelvic and paraortic lymph node dissection; RO: Right oophorectomy; WT: weight; *: 38 to 87 (median 68); **: unilateral adnexal mass: five patients; abdominal pain: six patients; abdominal distention: five patients; vaginal bleeding: three patients; nausea or vomiting: four patients; asymptomatic: one patient. ***: 11 to 22 cm (median 14.8 cm). ****: Ascites in one patient; #: Ascites in one patient; ##: 6.5 to 25 cm in largest dimension, with a mean of 13.9 cm (stdev ± 6.5 cm).

**Table 2 cancers-16-01106-t002:** Detailed results of the MBT patients’ characteristics.

Characteristic	Measure
Number of studies	48
Case reports	33 (69%)
Case series	15 (31%)
Number of patients	115
Patient age (years)	59.0 ± 13.3 (min: 22, max: 87)
Tumor size (cm)	12.8 ± 5.6 (min: 2.5, max: 30)
CA-125	Median: 53.4, Q1: 15, Q3: 224, min: 4, max: 4073
Ascites	33 patients out of 115 (28.7%)
Side	Left: 36 (37.1%), Right: 45 (46.4%), Bilateral: 16 (16.5%)
Stage	I: 50%, II: 7%, III: 32%, IV: 11%
Adjuvant therapy	No adjuvant therapy: 30 (31.25%), Chemotherapy: 59 (61.46%), Radiotherapy: 3 (3.13%), Chemotherapy and Radiotherapy: 4 (4.17%)
Second line therapy	No: 68, Chemotherapy: 26, Radiotherapy: 6, Surgery: 5
Recurrence	No: 57 (55.88%), Yes: 45 (44.12%)
Time to recurrence (months)	25.47 ± 26.20, median: 13, Q1: 9, Q3: 36, min: 3, max: 116
Follow up time (months)	40.89 ± 37.04, median: 27.5, Q1–Q3: 12–59, min: 1, max: 173
Outcome	ANED: 54 (54%), AWD: 9 (9%), DICU: 1 (1%), DOC: 6 (6%), DOD: 30 (30%)

ANED: alive with no evidence of disease; AWD: alive with disease; DOC: died of other cause; DOD: died of disease; DICU: died in intensive care unit.

**Table 3 cancers-16-01106-t003:** Comparison of results between women with recurrence and no recurrence.

Characteristic	Recurrence (*N* = 45) Median (Q1–Q3) or *N* (%)	No Recurrence (*N* = 57) Median (Q1–Q3) or *N* (%)	*p*-Value
Age	58.5 (49–69)	60 (48–70)	0.90748
CA-125 (U/mL)	91.7 (43–273.4)	27 (13–184.2)	0.11637
CA-125 (normal level)	8/19 (29.63%)	19/35 (70.37%)	0.0522
Tumor size (cm)	11 (7.5–15)	13.9 (9.5–16.5)	0.14999
Ascites	16/42 (38.1%)	11/49 (22.45%)	0.1033
Side (Right/Left and Right)	16/30 (53.33%)	26/40 (65%)	0.3241
Stage I	11/45 (24.44%)	34/45 (75.56%)	0.0018
Stage II	19/31 (61.29%)	12/31 (38.71%)	
Stage III	5/7 (71.43%)	2/7 (28.57%)	
Stage IV	5/7 (71.43%)	2/7 (28.57%)	
Adjuvant therapy	4/42 (9.52%)	4/50 (8%)	0.7961

**Table 4 cancers-16-01106-t004:** Clinical and histologic features of malignant Brenner tumors and their differential diagnoses.

	History of Ca	Benign or Borderline BT	Bilaterality	Multinodular Architectural Pattern	Teratoma Component	Presence of Glands
MBT	Usually no	Present	Sometimes	No	Absent	Yes (*)
HGSC	Usually no	Absent	Sometimes	No	Absent	Yes (**)
Primary SCC	Usually no	Absent	No	No	Absent	No
SCC arising in MT	Usually no	Absent	No	No	Present	No
End-Ca	Usually no	Absent	No	No	Absent	Yes (***)
Metastatic SCC	Yes	Absent	Yes	Yes	Absent	No
Metastatic UCa	Yes	Absent	Yes	Yes	Absent	No

BT: Brenner tumor; Ca: carcinoma; End-Ca: endometrioid carcinoma; MBT: malignant Brenner tumor; MT: mature teratoma; HGSC: high-grade serous carcinoma; SCC: squamous cell carcinoma; UCa: urothelial carcinoma; *: mucinous glands; **: high grade cytological features; ***: endometrioid glands.

**Table 5 cancers-16-01106-t005:** Immunohistochemical features of malignant Brenner tumors and their differential diagnoses.

	WT-1	ER	GATA-3	p63
MBT	Negative	Negative	Positive	Positive
HGSC	Positive	Positive	Negative	Negative
Primary SqCC	Negative	Negative	Negative	Positive
SqCC arising in MT	Negative	Negative	Negative	Positive
Endometrioid Ca	Negative	Positive	Negative	Negative
Metastatic SqCC	Negative	Negative	Negative	Positive
Metastatic UCa	Negative	Negative	Positive	Positive

Ca: carcinoma; MBT: malignant Brenner tumor; MT: mature teratoma; HGSC: high-grade serous carcinoma; SqCC: squamous cell carcinoma; UCa: urothelial carcinoma.

## Data Availability

Not applicable.

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
