# Peer review of "Malignant Brenner Tumor of the Ovary: A Systematic Review of the Literature"

_cancers, 2024, doi:10.3390/cancers16061106_

Round 1
Reviewer 1 Report
Comments and Suggestions for Authors
This is a systematic review of the English literature on Brenner tumors of the ovary. It is a highly descriptive review that does not lead to any substantial recommendation for clinical management. This article could be helpful to clinicians treating patients affected by these rare tumors, but I don’t think it can be of much interest to the multidisciplinary readership of this journal. It would be better suited in a more specialized journal focused on gynecological oncology. The article would be improved by addressing the following specific points:
1. In the second paragraph of the introduction, the authors state that Brenner tumors arise from the surface epithelium. This used to be the favored hypothesis for epithelial ovarian tumors in general. There is no evidence for this, and, in fact, the most common subtypes of ovarian epithelial tumors are no longer thought to be arising from this epithelium. Unless the authors come up with some novel hypothesis, it is best to simply say that the origin of these tumors is unknown.
2. In section 3.5, c.4965C>G does not follow the current conventional nomenclature rules for single base substitutions in cDNA.
3. In Table 3, the position of text quoting the P value of 0.0018 implies that it refers to all stages while it instead refers to stage 1 tumors. The text should be placed on the same line as “Stage 1”.
4. On Page 5, the authors state that CA125 was higher in women with recurrence, but the P value is above 0.05. At the most, they could state that there may be an association with CA125 but there was not enough statistical power to make a definitive statement about this.
5. It is not clear if the Illustration at the bottom of page 5 belongs to figure 3 or not. If it does not, then, it has no label, and no legend. If it does, then to figure legend is not accurate illustrate data for CA125, while the upper left data refers to bilaterality.
6. Section 3.5 States that the BRCA2 mutation was the only molecular finding. However, in the discussion, the authors point out to a variety of additional Bunnicula findings that were reported. This discrepancy needs to be resolved. In addition, the molecular findings presented in the discussion should belong to the results. In fact, and additional table summarizing all the money to findings would be of interest and improve the quality of the article.
7. Table 1 spans over 12 pages in the methodology section. It is also submitted as a supplementary Table. I think that a supplementary Table is a better way to include this information instead of including it in a Table in the main text.
Author Response
First of all, we would like to thank all reviewers for their constructive comments. Below is a point-by-point answer to the reviewers’ comments.
Reviewer 1
- In the second paragraph of the introduction, the authors state that Brenner tumors arise from the surface epithelium. This used to be the favored hypothesis for epithelial ovarian tumors in general. There is no evidence for this, and, in fact, the most common subtypes of ovarian epithelial tumors are no longer thought to be arising from this epithelium. Unless the authors come up with some novel hypothesis, it is best to simply say that the origin of these tumors is unknown.
Answer: We thank the reviewer for this allowing us to clarify this issue. We have changed the second paragraph of the introduction section as required. We have modified the manuscript based on the content of the latest WHO edition of female genital tumors.
- In section 3.5, c.4965C>G does not follow the current conventional nomenclature rules for single base substitutions in cDNA.
Answer: We have erased c.4965C>G from the section 3.5
- In Table 3, the position of text quoting the P value of 0.0018 implies that it refers to all stages while it instead refers to stage 1 tumors. The text should be placed on the same line as “Stage 1”.
Answer: We have corrected as advised.
- On Page 5, the authors state that CA125 was higher in women with recurrence, but the P value is above 0.05. At the most, they could state that there may be an association with CA125, but there was not enough statistical power to make a definitive statement about this.
Answer: We have added the comment as advised.
- It is not clear if the Illustration at the bottom of page 5 belongs to figure 3 or not. If it does not, then it has no label and no legend. If it does, then to figure legend is not accurate illustrate data for CA125, while the upper left data refers to bilaterality.
Answer: We thank the reviewer for bringing this issue to our attention. We have modified it accordingly. In addition, the legend is revised to depict the actual content of the subfigures.
- Section 3.5 States that the BRCA2 mutation was the only molecular finding. However, in the discussion, the authors point out to a variety of additional Bunnicula findings that were reported. This discrepancy needs to be resolved. In addition, the molecular findings presented in the discussion should belong to the results. In fact, and additional table summarizing all the money to findings would be of interest and improve the quality of the article.
Answer: We thank the reviewer for allowing us to clarify this issue. BRCA2 was indeed the only molecular finding reported in the cases we reviewed. The remaining findings were included in other manuscripts that did not meet the inclusion criteria for the systematic analysis. For this reason, they were included in the discussion section and not in section 3.5.
- Table 1 spans over 12 pages in the methodology section. It is also submitted as a supplementary Table. I think that a supplementary Table is a better way to include this information instead of including it in a Table in the main text.
Answer: Supplementary Table 1 provides more details concerning treatment. We believe that we should keep Table 1 in the main text since it provides valuable information.
Reviewer 2 Report
Comments and Suggestions for Authors
(please see the attached file)

Author Response
First of all, we would like to thank all reviewers for their constructive comments. Below is a point-by-point answer to the reviewers’ comments.
Reviewer 2.
- What are the differences in the clinical presentation at diagnosis (e. age, stage, symptoms, e.t.c.) between MBT and overall epithelial ovarian cancer?
Answer:
- We have described the differences in the clinical presentation at diagnosis between MBT and epithelial ovarian cancer
- What are the differences in the survival outcomes (e. progression free survival, overall survival, pattern of recurrence) between MBT and overall epithelial ovarian cancer?
Answer: It is commonly accepted that epithelial ovarian cancer is a very heterogeneous disease with diverse pathogenesis, molecular features, clinical course, and treatment. For example, high-grade serous ovarian cancer is regarded as a chemosensitive disease compared to low-grade one, and complete cytoreduction is the cornerstone treatment. However, even among grade 3 serous tumors those high GIS carry a more favorable prognosis compared to HRP tumors. Besides, MBT is commonly regarded as an aggressive malignancy with only little similarity to the more frequent high-grade serous. Due to the rarity of this entity, the absence of driving mutations, the lack of homogeneous treatment guidelines, and its little similarity with the other epithelial tumors, we cannot offer comparative data of prognosis and survival.
- According to the data in this study, should we use more aggressive or more conservative treatment for MBT?
Answer: We thank the reviewer for giving us the opportunity to clarify this issue. Although it is not prospectively studied, our retrospective analysis showed that lymph node dissection is associated with better prognosis. However, this finding should be prospectively validated. Until then, it is accepted that BMT should be managed as a common epithelial ovarian cancer with the goal of achieving complete cytoreduction. We have included this part in the Discussion.
- Since “patients with lymph node dissection (had better survival than patients without L ND p 0.0131) should thorough lymphadenectomy be recommended for MBT? Was the role of lymphadenectomy more likely to be therapeutic, or only diagnostic to achieve a more accurate staging?
Answer: Although it is not prospectively studied, our retrospective analysis showed that lymph node dissection is associated with a better prognosis. However, this observational finding does not provide a possible causal association between the prognostic or therapeutic value of the lymph node dissection. Although the LION study showed that the dissection of enlarged-abnormal lymph nodes adds therapeutical value in patients with advanced disease,
Round 2
Reviewer 1 Report
Comments and Suggestions for Authors
The authors adequately addressed comments from my previous review
Reviewer 2 Report
Comments and Suggestions for Authors
This is a detailed review of literature about malignant Brenner tumor of the ovary.